# Non-invasive smooth muscle electromyography (SMEMG) as a novel monitoring technology of the gastrointestinal tract of awake, free-moving pigs—A pilot study

Katalin Nagy[1], Hedvig Fébel[2], George Bazar[1,3]*, György Grosz[4], Róbert Gáspár[5], Kálmán Ferenc Szűcs[5], Tamás Tóth[1,3]

1 Institute of Animal Physiology and Nutrition, Hungarian University of Agricultural and Life Sciences, Kaposvár, Hungary, 2 Institute of Animal Physiology and Nutrition, Hungarian University of Agricultural and Life Sciences, Herceghalom, Hungary, 3 ADEXGO Kft., Balatonfüred, Hungary, 4 MSB-MET Kft., Balatonfüred, Hungary, 5 Department of Pharmacology and Pharmacotherapy, Faculty of Medicine, University of Szeged, Szeged, Hungary

* bazar@agrilab.hu

**Data Availability Statement:** All relevant data are within the paper and its Supporting Information files.

## Abstract

There are several mathematical models and measurements to determine the efficiency of the digestibility of different feedstuffs. However, there is lack of information regarding the direct methods or measurement techniques used to analyse the physical response of the different parts of the gastrointestinal tract (GIT) of growing pigs to different diets. Smooth muscle electromyography (SMEMG) is a non-invasive method for the measurement of gastrointestinal myoelectrical activity. In the present study, SMEMG methodology has been adapted from laboratory rats to pigs, and the effects of feedstuffs with control (CTR) or experimentally increased (EXP) amounts of fibre were investigated on gastrointestinal tract motility. Nine barrow pigs ((Danish Landrace × Danish Yorkshire) × Danish Duroc) were used (30 ± 3 kg), and their CTR and EXP feedstuffs contained 29 and 49 g/kg crude fibre (CF), respectively. Myoelectric activities of the stomach, ileum and caecum were detected in the awake pigs by a pair of electrodes. The recorded myoelectric signals were analysed with fast Fourier transformation (FFT), and the spectra were expressed in GIT section-specific cycles per minutes (cpm) values and the maximum power spectrum density ($PsD_{max}$). A significant increase ($P < 0.001$) was observed in the value of the $PsD_{max}$ of the small intestine (20–25 cpm) as a consequence of the EXP diet. The PsDmax values of the stomach (3–5 cpm) and large intestine (1–3 cpm) did not show any significant change in pigs fed the EXP diet. As a direct and non-invasive method, SMEMG is suitable for the rapid evaluation of the effects of diets with different fibre contents on the GIT of non-anaesthetised, free-moving pigs.

**Funding:** This publication was prepared within the framework of the predoctoral scholarship program of the Ministry of Human Capacities, Hungary (project number: EFOP-3.6.3-VEKOP-16-2017-00005). The results discussed in this publication are the outcome of an independent research that was part of a PhD work. The conditions (stable, animals, human resources) were provided and financed by Széchenyi István University and Hungarian University of Agricultural and Life Sciences, which has been recognized in the „Acknowledgements" section. Commercial R&D companies, ADEXGO Kft. and MSB-MET Kft. provided support in the form of salaries for authors [G. Bazar, Gy. Grosz, T. Tóth], but did not have any additional role in the study design, data collection and analysis, decision to publish, or preparation of the manuscript. The specific roles of these authors are articulated in the 'author contributions' section.

**Competing interests:** Authors G. Bazar and T. Tóth are partly employed by a commercial R&D company, ADEXGO Kft., and Gy. Grosz is employed by a commercial R&D company, MSB-MET Kft. However, these conditions do not alter the authors' adherence to PLOS ONE policies on sharing data and materials. The mentioned companies only provided financial support in the form of authors' salaries, and did not play a role in the study design, data collection and analysis, decision to publish, or preparation of the manuscript.

## Introduction

Digestion is affected by many factors, such as the physical and chemical characteristics of feed [1], feed processing [2], different animal factors and nutritional levels [3]. Dietary fibre regulates motility and digestive transit time through the gastrointestinal tract (GIT) [4].

There are several models to predict the digestibility of different feedstuffs. These models can integrate theories and observations to obtain a comprehensive view of complex biological systems [5]. Ileal or total tract digestion models have been developed for pigs [6–8], in which digestibility is predicted by separately quantifying transit time, endogenous secretions, degradation, absorption and microbial fermentation. These predictions are required because of the limited quantitative information concerning the digestion and transit kinetics in the different segments of the GIT of pigs. Moreover, the gastric emptying process is represented by different masses of feedstuff or segments of the digestive tract [6–8].

Only a few methods are available to analyse the physical reaction of the different parts of the GIT to the consumed feed. Electrogastrography (EGG) is a non-invasive method for the measurement of gastric myoelectrical activity [9]. The GIT smooth muscle has pacemaker cells, called the interstitial cells of Cajal (ICCs), that play a key role in the generation and propagation of the electric signal in GIT contractility, which is measurable. The newly developed smooth muscle electromyography (SMEMG) method examines the different organs of the GIT (the stomach, small and large intestine), rather than examining only gastric myoelectrical activity as in the case of EGG. Pigs can be used in various preclinical experiments [10–15] as an omnivorous representative due to their relatively similar gastrointestinal functions to those of humans [16, 17].

In human (especially awake) patients and large model animals such as pigs, there are few possibilities to record myoelectric signals directly from the colon or small intestine other than using electrodes placed on the body surface, as the placement of various internal sensors is considerably difficult. However, in a rat experiment [18] electrodes were implanted on the outer surface of the digestive tract, and the slow-wave frequency parameters of the gastric, small intestine and large intestine segments of the GIT were identified using smooth muscle electromyography. The changes in the myoelectric activity of the GIT were followed with SMEMG in parallel with the mechanical contractions in anaesthetised rats using a strain gauge tool. Another SMEMG experiment was carried out in awake rats, measuring GIT activity under normal and stressed conditions [19].

Based on these earlier findings, our study primarily aims to develop the non-invasive SMEMG measurement applied for pigs, and to identify and analyse myoelectric signals of the different GIT segments in growing and awake pigs. The secondary objective is to investigate the effect of fibre-rich nutrition on the GIT activity of pigs, using the applied SMEMG method.

## Materials and methods

### Housing and handling of the animals

The animals were treated following the European Communities Council Directives (2010/63/EU) and the Hungarian Act for the Protection of Animals in Research (Article 32 of Act XXVIII). All experiments involving animal subjects were carried out with the approval of the Hungarian Ethical Committee for Animal Research (registration number: VIII-I-001/01854-0005/2014).

Nine barrow pigs ((Danish Landrace × Danish Yorkshire) × Danish Duroc) were used at age 72 ± 3 days. Average body weight (BW) was 30 ± 3 kg. The animals were housed

| Adaptation period 1 CTR diet | **Measuring day** | Adaptation period 2 EXP diet | **Measuring day** | Adaptation period 3 CTR diet | **Measuring day** | Adaptation period 4 EXP diet | **Measuring day** |
|---|---|---|---|---|---|---|---|
| 5 days | **7.00h** | 5 days | **7.00h** | 5 days | **12.00h** | 5 days | **12.00h** |

**Fig 1. The experimental design for measuring the gastrointestinal myoelectric activity (n = 9 pigs).** The diet of measuring days was identical to that of the preceding 5 days of adaptation. CTR: low level of crude fibre (29 g/kg as in feed), soybean- and maize-based feed; EXP: moderate level of crude fibre (49 g/kg as in feed), soybean- and maize-based feed + 4% Opticell (Agromed Austria GmbH, Kremsmünster, Austria).

individually (1 pig/pen, 1 m$^2$/pig) without litter, and water was offered ad libitum. Fig 1 summarises the full random crossover trial design, in which the two diets (control and experimental) were administered in an alternating manner in four iterative cycles, each comprising an adaptation period (five days) and measuring day.

## Diet and chemical analysis

The control diet (CTR) contained a low level of crude fibre (CF; 29 g/kg as in feed) and was prepared from wheat, maize, barley and soybean meal without a specific fibre source. The experimental diet (EXP) corresponded to the control diet, except that it had an increased CF content (49 g/kg as in feed) due to the addition of a concentrated fibre source (Opticell C5, Agromed Austria GmbH, Kremsmünster, Austria). Ingredients and analysed or calculated chemical characteristics of the diets are given in Table 1.

Dry matter (DM), crude protein (CP), crude fibre (CF), neutral detergent fibre (NDF), acid detergent fibre (ADF), acid detergent lignin (ADL), ether extract (EE), ash (A) of CTR and EXP diets were determined. The chemical analysis of the diets was performed following the AOAC protocol [20] and the standard laboratory procedures.

Outside the experimental period, pigs received the CTR diet. The control and experimental diets were fed for five days before the SMEMG measurements were taken. The animals received the daily amount of feed in two equal portions, covering 2.8 times their maintenance energy requirement (450 kJ $ME_s$/kg$^{0.75}$/day). The daily amount of feed was calculated based on the body weight (BW) of the animals (the growing pigs were weighed every week, and the fed diet was adjusted to their BW). The pigs were fed at 7.00 a.m. and 12.00 a.m.

## Detection of gastrointestinal myoelectric activity

SMEMG measurements were performed on the "measuring days" (Fig 1), starting at the time of feeding (7.00 a.m. or 12.00 a.m.), and lasted for 4 hours per animal. Before the measurement, the epigastric area was cleaned, and the skin was gently shaved. The electrodes (Electrode PE Foam Solidgel, Bio Lead-Lok B Sp. Zo.o, Józefów, Poland) were fixed onto the surface of the skin with adhesive plaster (Leukoplast 5 cm, BSN medical GmbH, Hamburg, Germany), without surgery. Ten20 EEG conductive gel (Bio-Medical Instruments, USA) was used on the skin surface to provide proper conductivity of the electrodes. The standard electrode pairs (2 electrodes) were fixed to the right and left sides of the abdominal wall, 10 cm lateral from the spine, while the neutral electrode was placed on the right thigh at a distance of 5 cm from the tail. A custom belt was designed to fix and protect the electrodes and hold the halter device (MDE GmbH, Walldorf, Germany), which recorded and stored the myoelectric signals (Fig 2).

The slow-wave frequency parameters of the gastric, small intestine and large intestine segments of the GIT in rats identified by Szűcs et al. [18] were used in this study. The electric signals were recorded and analysed with an on-line computer and amplifier of the S.P.E.L.

**Table 1. Ingredients and the analysed or calculated nutrient and energy content of the control and experimental diets.**

| Ingredient (kg) | Control (CTR) | Experimental (EXP) |
|---|---|---|
| Wheat | 307.70 | 295.40 |
| Maize | 256.50 | 247.00 |
| Barley | 200.00 | 192.00 |
| Extracted soybean meal (46% CP) | 188.00 | 181.00 |
| Pig vitamin and mineral premix[1] | 12.50 | 12.50 |
| Plant oil[2] | 12.00 | 11.20 |
| Limestone | 10.50 | 10.05 |
| MCP | 6.50 | 6.20 |
| L-lysine-HCl (78%) | 4.80 | 4.60 |
| DL-methionine (99%) | 0.64 | 0.62 |
| L-threonine (99%) | 0.64 | 0.62 |
| L-tryptophan (98.5%) | 0.22 | 0.21 |
| Fibre source[3] | - | 38.60 |
| **Chemical composition (as in feed, %)** | | |
| DM[4] | 88.90 | 89.10 |
| CP[4] | 17.20 | 16.70 |
| EE[4] | 3.30 | 3.30 |
| CF[4] | 2.90 | 4.90 |
| NDF[4] | 11.86 | 14.70 |
| ADF[4] | 4.89 | 7.50 |
| ADL[4] | 0.73 | 0.70 |
| A[4] | 4.20 | 4.30 |
| Ca[5] | 0.66 | 0.64 |
| P[5] | 0.55 | 0.48 |
| Na[5] | 0.20 | 0.20 |
| SID Lys[5] | 1.05 | 1.01 |
| SID Met+Cys[5] | 0.66 | 0.64 |
| SID Thr[5] | 0.70 | 0.68 |
| SID Trp[5] | 0.21 | 0.20 |
| $DE_s$[5] (MJ/kg as in feed) | 14.17 | 13.67 |
| $ME_s$[5] (MJ/kg as in feed) | 13.60 | 13.12 |

[1]Supplied per kilogram of diet: vitamin A: 16,000 IU; vitamin D: 2500 IU; vitamin E: 80.0 mg; vitamin K: 2.0 mg; thiamine: 2.0 mg; riboflavin: 6.0 mg; niacin: 30.0 mg; pantothenic acid: 14.0 mg; pyridoxine: 4.0 mg; folic acid: 1.0 mg; vitamin B12: 0.03 mg; I: 0.7 mg (calcium iodate); Se: 0.4 mg (sodium selenite); Zn: 120.0 mg (metal polysaccharide complexes of zinc sulphate); Fe: 90.0 mg (iron(II)-carbonate); Mn: 50.0 mg (manganese sulphate); Cu: 95.0 mg (copper sulphate).

[2]Bonafarm-Bábolna Ltd., Nagyigmánd, Hungary.

[3]Opticell C5 concentrated fibre source: crude fibre: 660 g/kg DM, NDF: 854 g/kg DM, ADF: 725 g/kg DM, ADL: 82 g/kg DM (Agromed Austria GmbH, Kremsmünster, Austria).

[4]analysed (DM—dry matter; CP—crude protein; EE—ether extract; CF—crude fibre; NDF—neutral detergent fibre; ADF—acid detergent fibre; ADL—acid detergent lignin; A = ash).

[5]calculated (Ca—calcium; P—total phosphorous; Na—sodium; SID Lys—standardised ileal digestible lysine; SID Met + Cys—standardised ileal digestible methionine + cysteine, SID Thr—standardised ileal digestible threonine; SID Trp —standardised ileal digestible tryptophan; $DE_s$—digestible energy for swine; $ME_s$—metabolisable energy for swine).

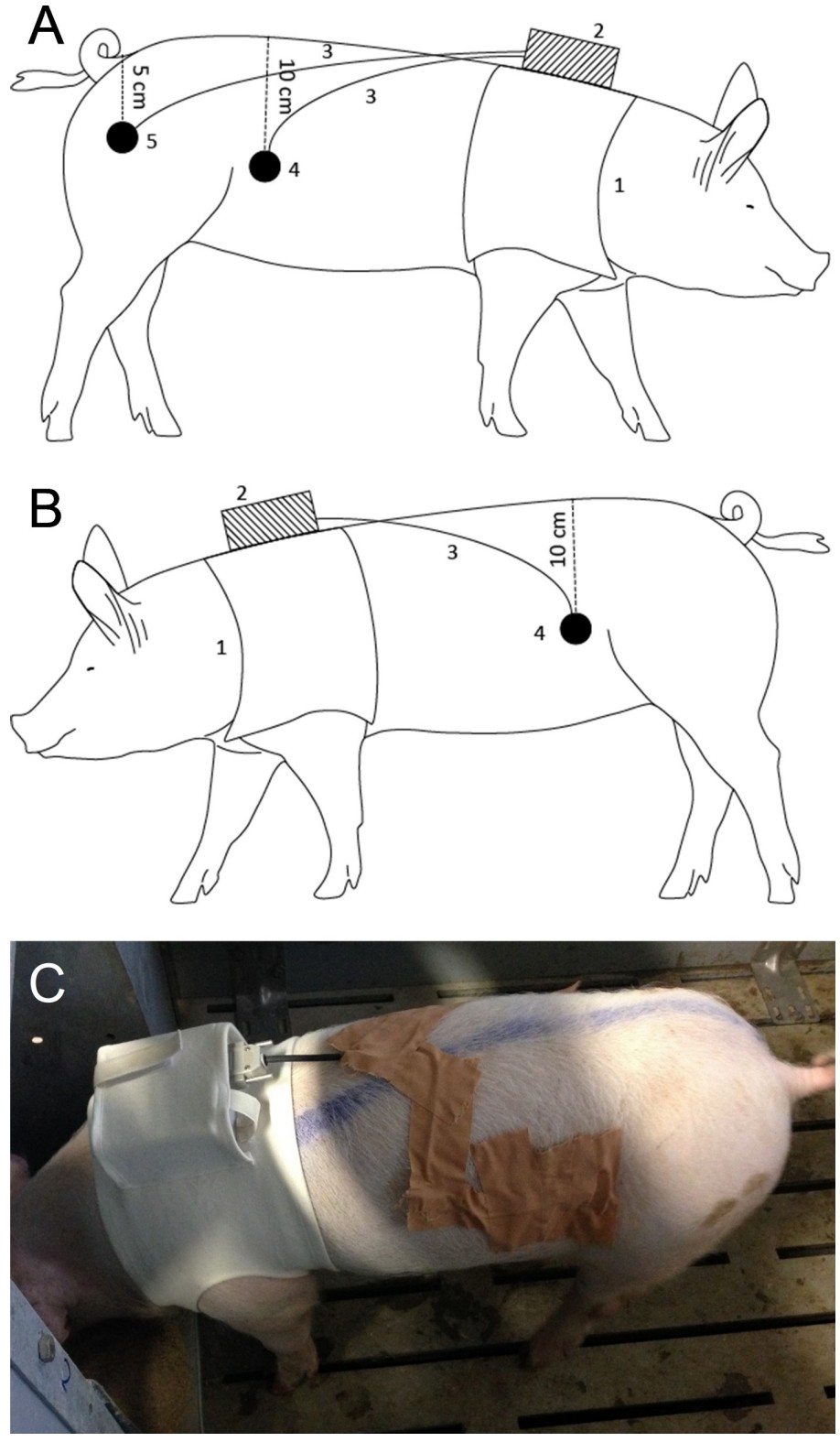

**Fig 2.** The positioning of the smooth muscle electromyography equipment on the right (A) and left (B) side of a standing animal, and a photo of the left side (C). 1: belt; 2: halter part; 3: cable for the electrode(s); 4: standard electrode; 5: neutral electrode.

Advanced ISOSYS Data Acquisition System (MDE GmbH, Walldorf, Germany). SMEMG signals were amplified using a custom-made amplifier (MDE GmbH, Walldorf, Germany). A double-filter system was used to reduce the artefacts. All analogue signals were pre-filtered with a first-order Bessel-type low-pass filter and were converted to digital signals at a sample rate of 2 Hz with a slope of 80 dB/decade. The prefiltered myoelectric signals were then filtered further by Bessel-type bandpass filters with a frequency of 0–30 cycles per minute with a slope of 140 dB/decade. Each filter was a digital infinite impulse response (IIR) filter. The recorded signals were analysed with fast Fourier transformation (FFT). The frequency of the electric activity was expressed in cycles per minutes (cpm), and the magnitude of the activity was described as the maximum power spectrum density ($PsD_{max}$). Based on Szűcs et al. [18], the electromyographic signals during FFT analysis were filtered for the following cpms: 1–3 for the large intestine, 3–5 for the stomach and 20–25 for the small intestine.

## Statistical analysis

The maximum power spectrum density ($PsD_{max}$) values were determined and compared statistically (*t*-test) using the computer program Prism 5.0. (GraphPad Software, USA). In all procedures, the significance level was $P \leq 0.05$.

## Results and discussion

The activities of the different GI segments were detected by FFT spectrum analysis of the SMEMG records (S1 File, Fig 3A). The peaks of 1–3, 3–5 and 20–25 cpms refer to the contractile responses of the large intestine, stomach and small intestine, respectively (Fig 3B). The absolute values of $PsD_{max}$ ($mV^2$) varied in the different segments of the GIT (the stomach, small intestine and large intestine) and the different feeds (CTR and EXP), indicating changes in the intensity of the GIT smooth muscle contractions. Table 2 shows the results of mean values of the maximum of power spectrum density (PsDmax, mV2) changes for each individual animal (pig 1–9). The means and standard deviations of the PsDmax values of the control and experimental animals (n = 9) examined in the test are shown in Fig 4.

The $PsD_{max}$ measured when feeding the EXP diet differed from that measured when feeding the CTR diet by the individual animals. These changes were neither consistent nor significant in the stomach or large intestine. The $PsD_{max}$ ($mV^2$) when feeding the CTR feed varied between 101.3 (animal 6) and 350.3 (animal 1) for the stomach, between 20.4 (animal 5) and 67.2 (animal 4) for the small intestine, and between 44.0 (animal 7) and 362.9 (animal 4) for the large intestine. As a result of the EXP feed, the $PsD_{max}$ ($mV^2$) varied between 102.8 (animal 4) and 355.6 (animal 9) for the stomach, between 26.4 (animal 4) and 174.2 (animal 7) for the small intestine, and between 64.8 (animal 4) and 385.8 (animal 5) for the large intestine. A significant increase (CTR vs. EXP) in the $PsD_{max}$ values were found only for the small intestine in the case of all individuals (n = 9) ($P \leq 0.001$) (Fig 4).

Earlier experiments demonstrated that SMEMG is reliable and feasible for use in anaesthetised pigs, especially for examination of the stomach (EGG) [10, 21, 22]. Although animal SMEMG methods are promising, it cannot be guaranteed that the recorded signals originate solely from the GI tracts. The signals contain slow and fast waves of the GIT along with "noises", such as respiratory and motion artefacts [23]. Many studies have highlighted that the frequencies outside the normal frequency are defined as a type of arrhythmia [24]. The possibility of slow-wave myoelectric signal interference, or even masking with fast-wave signals from the brain, cardiac muscle or skeletal muscle, is very high, but there is an opportunity to reduce this through the special design of the sensors [25]. There are no unequivocal data concerning the frequency parameters of specific myoelectric signals of the main segments of the

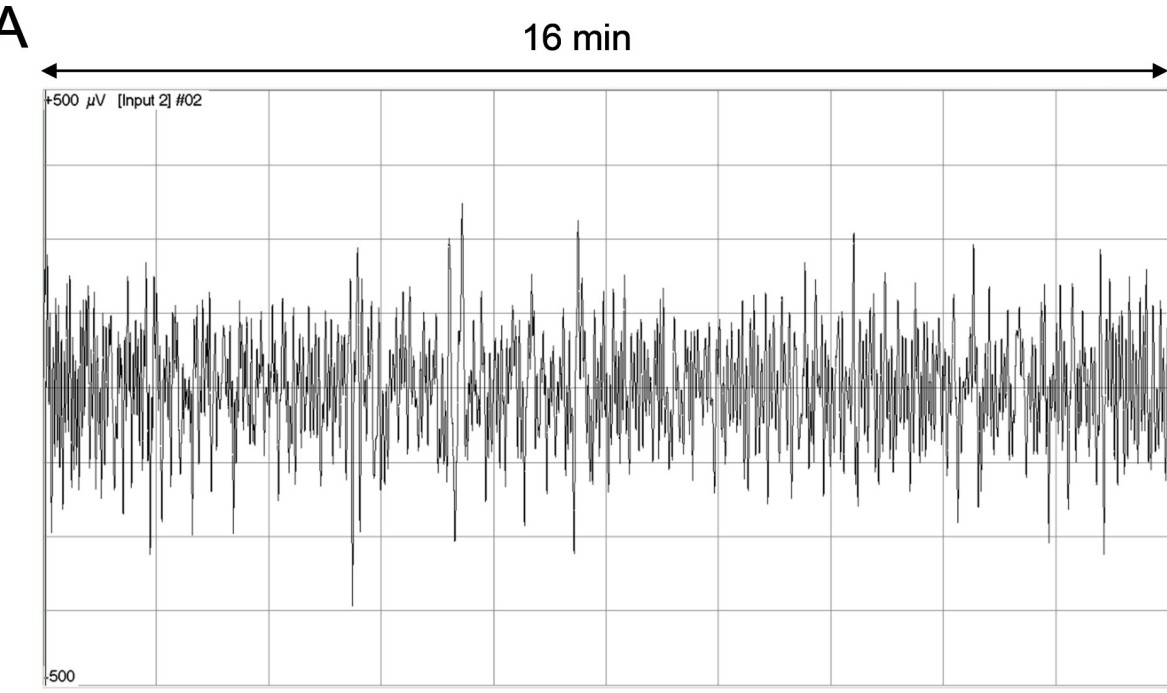

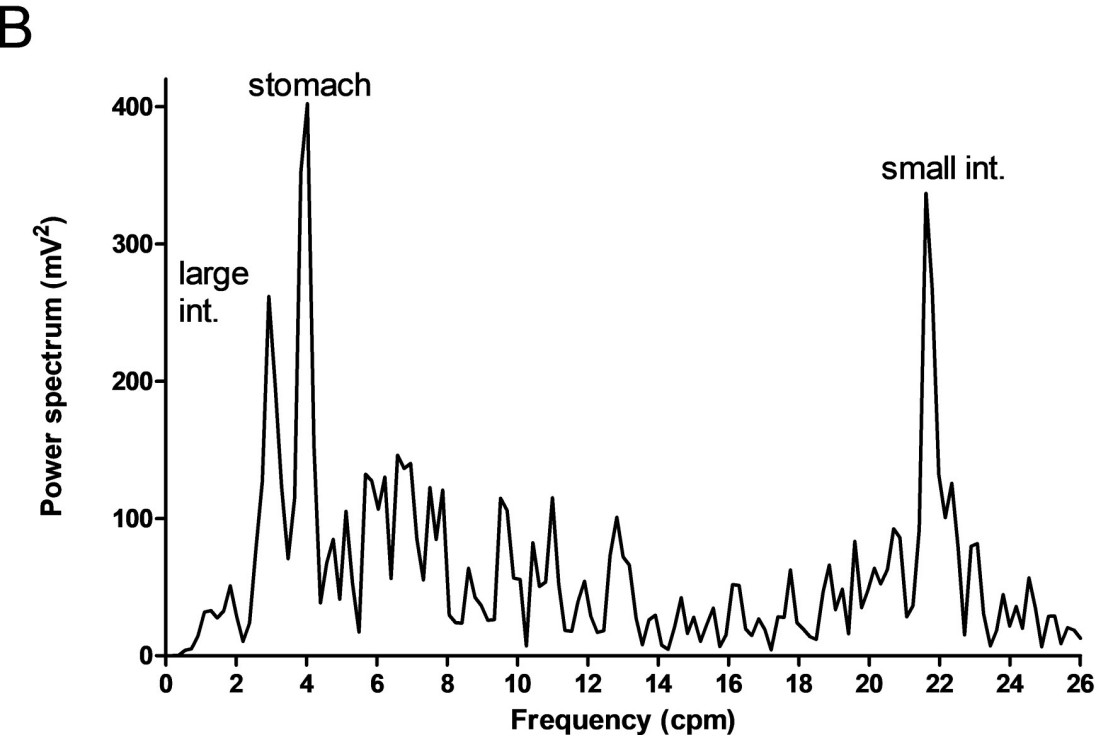

**Fig 3.** The primary myoelectric signal of the gastrointestinal tract in a non-anaesthetised, free-moving pig (A), and the specific spectra (B) gained by fast Fourier transformation (FFT) showing three dominant peaks particularly notable between 1–3, 3–5 and 20–25 cpms for the large intestine, stomach and small intestine, respectively. The heights of the peaks are the $PsD_{max}$ values that were compared to characterise the changes in the intensities of the given GIT segments caused by the different fibre contents of the feeds.

**Table 2. The effect of the control and experimental diets on the absolute value of the maximum of power spectrum density (PsDmax, mV2) in the different segments of the gastrointestinal tract (n = 9 pigs).**

| | Absolute value for $PsD_{max}$ ($mV^2$) | | | | | |
|---|---|---|---|---|---|---|
| | Stomach (3–5 cpm) | | Small intestine (20–25 cpm) | | Large intestine (1–3 cpm) | |
| Animal No. | CTR | EXP | CTR | EXP | CTR | EXP |
| 1 | 350.3 | 349.0 | 66.2[b] | 153.7[a] | 211.5 | 249.5 |
| 2 | 191.3 | 174.4 | 43.2[b] | 163.6[a] | 86.6 | 140.7 |
| 3 | 269.8 | 223.3 | 22.6[b] | 133.0[a] | 178.2 | 126.5 |
| 4 | 306.6 | 102.8 | 67.2[a] | 26.4[b] | 362.9 | 64.8 |
| 5 | 208.3 | 169.2 | 20.4[b] | 168.8[a] | 56.6 | 385.8 |
| 6 | 101.3 | 182.3 | 22.3[b] | 121.3[a] | 137.4 | 120.6 |
| 7 | 116.5 | 106.4 | 20.5[b] | 174.2[a] | 44.0 | 89.9 |
| 8 | 225.2 | 196.0 | 65.9[b] | 92.6[a] | 179.9 | 123.6 |
| 9 | 250.2 | 355.6 | 66.5[b] | 82.9[a] | 84.0 | 178.2 |

Values with different superscripts differ at $P \leq 0.001$.

CTR: low level of crude fibre (29 g/kg as in feed), soybean- and maize-based feed.

EXP: moderate level of crude fibre (49 g/kg as in feed), soybean- and maize-based feed + 4% Opticell (Agromed Austria GmbH, Kremsmünster, Austria).

GIT (the stomach, small and large intestine) for pigs; however, Tacheci et al. [22] defined the normal range for gastric myoelectric activity in anesthetised experimental pigs at 2.3–3.5 cpm. Varayil et al. [21] also worked with anesthetised pigs but in fasting conditions and measured normal dominant frequencies for the empty stomach at 3.3 ± 0.5 cpm. The identified dominant cpms of each experiment were inside the cpm value range of the stomach obtained in our experiment (3.00–5.00 cpm). These values were in accordance with earlier findings in humans [26–28].

In our recent study, the same recording and analysis software as those used by Szűcs et al. [18] in rats were adopted together with digital filters to separate the slow waves of smooth

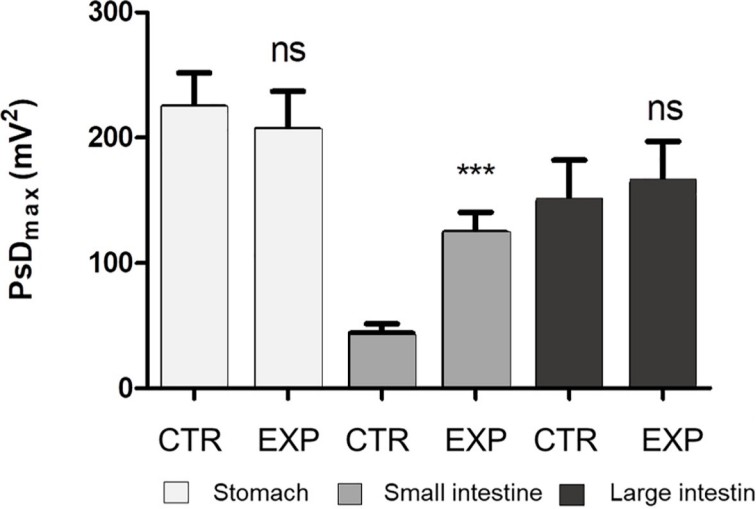

**Fig 4. The differences of the absolute values of the maximum of power spectrum density (PsD$_{max}$, mV$^2$) for the stomach, small intestine and large intestine caused by being fed CTR or EXP diets (n = 9 pigs).** ns: non-significant difference between the PsD$_{max}$ values of the CTR and EXP feeding periods; ***: significant ($P < 0.001$) difference between the PsD$_{max}$ values of the CTR and EXP feeding periods; CTR: low level of crude fibre (29 g/kg as in feed), soybean- and maize-based feed; EXP: moderate level of crude fibre (49 g/kg as in feed), soybean- and maize-based feed + 4% Opticell (Agromed Austria GmbH, Kremsmünster, Austria).

muscle signals from the cardiac, brain and skeletal muscle. This method allows the separation and characterisation of signals from the various sections of the GIT, as well as the detection of changes in smooth muscle activity.

To the best of our knowledge, there is no previous publication on SMEMG measurements performed in free-moving, awake pigs. The applicability and performance of the applied technology were tested during feeding diets with different concentrations of crude fibre. While fibre-induced motility and transit time changes have been widely studied, the effect of the fibre content of a diet on the transit time in the small intestine is not clear. Wenk et al. [29] suggested that fibre stimulates peristaltic action, and that it also reduces the transit time in the small intestine. Similarly, as a result of being fed a high-fibre diet, the presence of bulk exerted a direct physical action in the small and large intestine, which increased the peristaltic action and stimulated propulsive colonic motility, as reported by Laplace [30] using cannulated animals fed a diet with markers. Jorgensen et al. [31] reported a five- to six-fold increase in the digestive flow through the terminal ileum of pigs fed a high-fibre diet. On the other hand, van Leeuwen et al. [32] did not observe any effect of fibre on GIT activity or on the results of transit time in the small intestine.

In our study, the rat SMEMG measurement parameters were translated to pigs, and $PsD_{max}$ values for different parts of GIT were collected. The change in the maximum power spectrum density ($PsD_{max}$) of the stomach and small and large intestine was measured, which represents the change in the myoelectrical activity of the different segments of the GIT. A significant increase ($P < 0.001$) was observed in the absolute value of the $PsD_{max}$ of the small intestine as a consequence of the elevated dietary CF level. Based on the $PsD_{max}$ values, the contractions of the small intestine smooth muscle were more intense when the EXP diet was fed than when the CTR diet was fed. Such a change in the motility may have an impact on the transit time; thus, the results are in accordance with previous findings on the effect of increased fibre content in pig diets [30, 31].

## Conclusions

A durable SMEMG device was successfully tested as a portable tool for the non-invasive monitoring of gastrointestinal myoelectrical activity in awake, free-moving, growing pigs. The method is proven to be useful for the in vivo measurement of the altered myoelectric activity of the stomach and small and large intestine as a result of feeding diets with different fibre contents. The increased fibre content of the feed induced an increase in the $PsD_{max}$, indicating stronger GIT motility, especially in the small intestine. The applied method is useful to detect the influence of different feedstuffs on the GIT functions of pigs without anaesthesia.

## Supporting information

**S1 File. Smooth muscle electromyography records of the stomach, small intestine and large intestine of the investigated pigs (n = 9) acquired during the whole experiment (Fig 1).** CTR: control, EXP: experimental, T1: beginning of analysed time interval, T2: end of analysed time interval, CPM: cycle per minute, PS: power spectrum.
(XLSX)

## Acknowledgments

The authors are grateful to the employees of Hungarian University of Agricultural and Life Sciences, Kaposvár Campus, Institute of Animal Physiology and Nutrition, Department of

Animal Nutrition, and Széchenyi University, Faculty of Agricultural and Food Science, Department of Animal Nutrition for their contributions and technical support.

## Author Contributions

**Conceptualization:** Katalin Nagy, Tamás Tóth.

**Data curation:** Katalin Nagy.

**Formal analysis:** Róbert Gáspár, Kálmán Ferenc Szűcs.

**Methodology:** Róbert Gáspár, Kálmán Ferenc Szűcs.

**Resources:** György Grosz, Tamás Tóth.

**Software:** György Grosz.

**Supervision:** Hedvig Fébel, Tamás Tóth.

**Validation:** Róbert Gáspár, Kálmán Ferenc Szűcs.

**Visualization:** George Bazar, Kálmán Ferenc Szűcs.

**Writing – original draft:** Katalin Nagy.

**Writing – review & editing:** Katalin Nagy, Hedvig Fébel, George Bazar, Róbert Gáspár, Kálmán Ferenc Szűcs, Tamás Tóth.

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
