## [Decision Letter · Decision Letter 0]

16 Jun 2021

PONE-D-21-03411

Non-invasive smooth muscle electromyography (SMEMG) as a novel monitoring technology of the gastrointestinal tract of awake, free-moving pigs – a pilot study

PLOS ONE

Dear Dr. George Bazar,

Thank you for submitting your manuscript to PLOS ONE. After careful consideration, we feel that it has merit but does not fully meet PLOS ONE’s publication criteria as it currently stands. Therefore, we invite you to submit a revised version of the manuscript that addresses the points raised during the review process.

We look forward to receiving your revised manuscript.

Kind regards,

Ewa Tomaszewska, DVM Ph.D

Academic Editor

PLOS ONE

Journal Requirements:

"The Authors have declared that no competing interest exist."

We note that one or more of the authors are employed by a commercial company: ADEXGO Kft, MSB-MET Kft.

2.1. Please provide an amended Funding Statement declaring this commercial affiliation, as well as a statement regarding the Role of Funders in your study. If the funding organization did not play a role in the study design, data collection and analysis, decision to publish, or preparation of the manuscript and only provided financial support in the form of authors' salaries and/or research materials, please review your statements relating to the author contributions, and ensure you have specifically and accurately indicated the role(s) that these authors had in your study. You can update author roles in the Author Contributions section of the online submission form.

2.2. Please also provide an updated Competing Interests Statement declaring this commercial affiliation along with any other relevant declarations relating to employment, consultancy, patents, products in development, or marketed products, etc.  

Reviewers' comments:

Reviewer's Responses to Questions

**Comments to the Author**

1. Is the manuscript technically sound, and do the data support the conclusions?

Reviewer #1: Partly

Reviewer #2: Partly

2. Has the statistical analysis been performed appropriately and rigorously? 

Reviewer #1: Yes

Reviewer #2: I Don't Know

3. Have the authors made all data underlying the findings in their manuscript fully available?

Reviewer #1: Yes

Reviewer #2: No

4. Is the manuscript presented in an intelligible fashion and written in standard English?

Reviewer #1: Yes

Reviewer #2: Yes

5. Review Comments to the Author

Reviewer #1: The gastrointestinal tract has multifold tasks of ingesting, processing, and assimilating nutrients and disposing of wastes at appropriate times.

These tasks are facilitated by several stereotypical motor patterns that build upon the intrinsic rhythmicity of the smooth muscles that generate phasic contractions in many regions of the gut.

Phasic contractions timed by the occurrence of slow waves provide the basis for motility patterns such as gastric peristalsis and segmentation.

The method used allowed for the assessment or measurement of the electrical activity of the muscles. However, she did not consider what the test result was about, what osbzar the obtained results were from, etc.

There is no information of a slow wave or electrical discharges.

It is not known at what stage of the MMC the analysis was performed and this is important.

A number of biochemical and endocrinological factors play an important role in the regulation of electrical activity, there is no post-postal research in this field.

No information is available on this.

Reviewer #2: The authors presented an interesting article analyse myoelectric signals of the different GIT segments in growing and awake pigs and investigate the effect of fibre-rich nutrition on the GIT activity of pigs. Authors used non-invasive method for the measurement myoelectrical activity of stomach, small intestine and large intestine.

After analyzing the manuscript, several questions were raised:

1. The authors report that “two diets (control and experimental) were administered in an alternating manner in four iterative cycles, each comprising an adaptation period (five days) and measuring day.” while in the table the results are presented as the absolute value for individual animals I think that authors estimated maximum of power spectrum density -minimum twice for each diet. I think that multiple measurements during the day (“measuring days” lasted for 4 hours per animal) make it possible to present the results as mean values for individual animals

2. The control diet (CTR) contained a low level of crude fibre (CF; 29 g/kg as in feed) while the experimental diet (EXP) had an increased CF content (49 g/kg as in feed). In table 1 there can be observed Fibre source - only for experimental diet on the level 38.60 and without fibre source for control diet. It seems that the sources of fiber in the control and experimental groups should be similar and differ only in the amount.

6. PLOS authors have the option to publish the peer review history of their article (what does this mean?). If published, this will include your full peer review and any attached files.

Reviewer #1: No

Reviewer #2: No

---

## [Author Response · Author response to Decision Letter 0]

16 Jul 2021

Response to Reviewers’ Comments and Suggestions

PONE-D-21-03411

Non-invasive smooth muscle electromyography (SMEMG) as a novel monitoring technology of the gastrointestinal tract of awake, free-moving pigs – a pilot study

PLOS ONE

The authors would like to thank both the Editor and the Reviewers for considering the manuscript. All comments and recommendations have been thankfully accepted and corrections have been made accordingly. The changes are highlighted with Track Changes in the revised version of the manuscript. Please find our specific replies below.

Reviewer #1: 

The method used allowed for the assessment or measurement of the electrical activity of the muscles. However, she did not consider what the test result was about, what osbzar the obtained results were from, etc.

Thank you for the comment of the Reviewer, however it is not clear for us what is the main point of this special note. The word “osbzar” is an obvious typo, but unfortunately, we cannot figure out the meaning of it. Regarding the consideration of the test results, we can say that the whole manuscript is about the explanation of the test results, so it is not clear for us what is the point here raised by the Reviewer. We kindly ask the Reviewer to give clear description about the concerns raised against our manuscript.

In the Introduction part of the manuscript, we described in detail the main objectives of the work and previous problems. 

There is no information of a slow wave or electrical discharges.

The slow wave response was revealed on Figure 3, the whole methodological background is already published in our earlier manuscripts that are all cited in our recent publications (Szűcs KF, Nagy A, Grosz G, Tiszai Z, Gáspár R. Correlation between slow-wave myoelectric signals and mechanical contractions in the gastrointestinal tract: Advanced electromyographic method in rats. J Pharmacol Toxicol Methods. 2016;82: 37–44. doi:10.1016/j.vascn.2016.07.005; Szűcs KF, Grosz G, Süle M, Sztojkov-Ivanov A, Ducza E, Márki A, et al. Detection of stress and the effects of central nervous system depressants by gastrointestinal smooth muscle electromyography in wakeful rats. Life Sci. 2018;205: 1–8. doi:10.1016/j.lfs.2018.05.015). From these citations the Reviewer can get a wider insight to this methodology, but we cannot involve these earlier results into the current manuscript to avoid plagiarism. Since we filter out the fast electrical discharges during our record (previously we proved that these fast responses do not have significant impact on the mechanical contraction forces), therefore the slow electrical discharges are recorded as slow waves, only.

It is not known at what stage of the MMC the analysis was performed and this is important.

During our experiments we did 4-hour periods of measurements from 7 a.m. or 12.00 p.m. Since the animals have not been fasted, the stages of the MMC have no significance during the detection. Our aim was to measure the fed-induced alterations in the GI tract after meals. It is obvious that the MMC may had a contribution to the shape and frequency of slow waves, but their stages cannot be detectable clearly as it could be during/after fasting. Additionally, the 4-hour periods of detection very probably include the whole period of an MMC, thus we got a sum response instead of response of given stages. We think that our methods have higher practical importance providing a better insight to the effect of different fed on the GI tract motility in pigs.

A number of biochemical and endocrinological factors play an important role in the regulation of electrical activity, there is no post-postal research in this field. No information is available on this.

Thank you for this comment and we deeply agree that there are lots of factor that may modify the electric activity of the GI smooth muscle. However, such biochemical and endocrinological studies were not in the main scope of the study. Looking at Table 1 it becomes obvious that the main difference between the control and the test diet is the fiber content: the control diet did not contain any additional fiber sources while the test diet had added fiber content (Opticell C5 concentrated fibre source: crude fibre: 660 g/kg DM, NDF: 854 g/kg DM, ADF: 725 g/kg DM, ADL: 82 g/kg DM, Agromed Austria GmbH, Kremsmünster, Austria). Since the participating barrow pigs were in a self-control study without significant endocrinological alterations, and the fiber itself has low impact on endocrinological and biochemical modifications (especially within a 5-day diet), therefore we do not think that the above mentioned factors should be measured in this special experiments.

Reviewer #2 

The authors presented an interesting article analyse myoelectric signals of the different GIT segments in growing and awake pigs and investigate the effect of fibre-rich nutrition on the GIT activity of pigs. Authors used non-invasive method for the measurement myoelectrical activity of stomach, small intestine and large intestine.

Thank you very much for this comment. 

The authors report that “two diets (control and experimental) were administered in an alternating manner in four iterative cycles, each comprising an adaptation period (five days) and measuring day.” while in the table the results are presented as the absolute value for individual animals I think that authors estimated maximum of power spectrum density -minimum twice for each diet. I think that multiple measurements during the day (“measuring days” lasted for 4 hours per animal) make it possible to present the results as mean values for individual animals.

Thank you for this comment and we would like to clarify this. In the Table 2 the presented results are mean values of the maximum of power spectrum density (PsDmax, mV2) changes for each individual animal (animal, 1, 2, …9). However, the Figure 4 shows the mean values of PsDmax of the control and experimental animals (n=9) examined in the test. The relevant section has been improved in the manuscript accordingly. 

The control diet (CTR) contained a low level of crude fibre (CF; 29 g/kg as in feed) while the experimental diet (EXP) had an increased CF content (49 g/kg as in feed). In table 1 there can be observed Fibre source - only for experimental diet on the level 38.60 and without fibre source for control diet. It seems that the sources of fiber in the control and experimental groups should be similar and differ only in the amount.

The Table 1 shows that the main difference between the control and the experimental diet is the fiber content (29 g/kg as in feed vs. 49 g/kg as in feed, respectively): the control diet did not contain any additional fiber sources while the experimental diet had added fiber content (named “Opticell C5” with crude fibre: 660 g/kg DM, NDF: 854 g/kg DM, ADF: 725 g/kg DM, ADL: 82 g/kg DM, producer: Agromed Austria GmbH, Kremsmünster, Austria). We chose this kind of fiber source in our experiment because, (1) it is widely used in pig compound feeds in Europe, and (2) the crude fiber and fiber fractions (NDF, ADF) content of the product is constant and guaranteed. The used fiber source is also free of mycotoxins and other contaminants (e.g. heavy metals). At the beginning of the experiment, the most important nutrient content of the raw materials, including the fiber source, was evaluated in a proximate analysis. With the help of this concentrated fiber source, we had the opportunity to significantly increase the fiber content of the basic (control) compound feed without any differences in the main nutrients (such as protein, ether extract, minerals, and ileal digestible amino acids content) and energy content. The pigs in the experiment consumed the control and/or experimental diets without any residual feed (and every week the fed diets were adjusted to their body weight). Therefore, the differences (the absolute value of the maximum of power spectrum density) obtained in the experiment are solely due to the increased fiber (plus NDF and ADF-cellulose and hemicellulose) content of the test diet. The relevant section in the Materials and Methods has been extended with this explanation.

We thank once again the reviewers for their valuable comments that helped to improve the quality of our manuscript.

---

## [Decision Letter · Decision Letter 1]

31 Aug 2021

Non-invasive smooth muscle electromyography (SMEMG) as a novel monitoring technology of the gastrointestinal tract of awake, free-moving pigs – a pilot study

PONE-D-21-03411R1

Dear Dr. George Bazar,

We’re pleased to inform you that your manuscript has been judged scientifically suitable for publication and will be formally accepted for publication once it meets all outstanding technical requirements.

Kind regards,

Ewa Tomaszewska, DVM Ph.D

Academic Editor

PLOS ONE

Additional Editor Comments (optional):

Reviewers' comments:

Reviewer's Responses to Questions

**Comments to the Author**

1. If the authors have adequately addressed your comments raised in a previous round of review and you feel that this manuscript is now acceptable for publication, you may indicate that here to bypass the “Comments to the Author” section, enter your conflict of interest statement in the “Confidential to Editor” section, and submit your "Accept" recommendation.

Reviewer #2: All comments have been addressed

2. Is the manuscript technically sound, and do the data support the conclusions?

Reviewer #2: Yes

3. Has the statistical analysis been performed appropriately and rigorously? 

Reviewer #2: Yes

4. Have the authors made all data underlying the findings in their manuscript fully available?

Reviewer #2: Yes

5. Is the manuscript presented in an intelligible fashion and written in standard English?

Reviewer #2: Yes

6. Review Comments to the Author

Reviewer #2: The authors responded to the reviewer's previous comments in a satisfactory manner. I consider the author's' explanations sufficient for both suggestions (the "measuring days" and the comparison between the control and experimental diets)

7. PLOS authors have the option to publish the peer review history of their article (what does this mean?). If published, this will include your full peer review and any attached files.

Reviewer #2: No

---

## [Editor Report · Acceptance letter]

3 Sep 2021

PONE-D-21-03411R1 

Non-invasive smooth muscle electromyography (SMEMG) as a novel monitoring technology of the gastrointestinal tract of awake, free-moving pigs – a pilot study 

Dear Dr. Bazar:

I'm pleased to inform you that your manuscript has been deemed suitable for publication in PLOS ONE. Congratulations! Your manuscript is now with our production department. 

Kind regards, 

on behalf of

Prof. Dr. Ewa Tomaszewska 

Academic Editor

PLOS ONE